Resource

# Islet Gene View—a tool to facilitate islet research

Olof Asplund[1,2], Petter Storm[1,2,3,*], Vikash Chandra[4,*], Gad Hatem[1,2], Emilia Ottosson-Laakso[1,2], Dina Mansour-Aly[1,2], Ulrika Krus[1,2], Hazem Ibrahim[4], Emma Ahlqvist[1,2], Tiinamaija Tuomi[1,2,5,6], Erik Renström[1,2], Olle Korsgren[7,8], Nils Wierup[1,2], Mark Ibberson[9], Michele Solimena[10,11,12], Piero Marchetti[13], Claes Wollheim[1,2,14], Isabella Artner[1,2], Hindrik Mulder[1,2], Ola Hansson[1,2,6], Timo Otonkoski[4,15], Leif Groop[1,2,6], Rashmi B Prasad[1,2,6,16]

**Characterization of gene expression in pancreatic islets and its alteration in type 2 diabetes (T2D) are vital in understanding islet function and T2D pathogenesis. We leveraged RNA sequencing and genome-wide genotyping in islets from 188 donors to create the Islet Gene View (IGW) platform to make this information easily accessible to the scientific community. Expression data were related to islet phenotypes, diabetes status, other islet-expressed genes, islet hormone-encoding genes and for expression in insulin target tissues. The IGW web application produces output graphs for a particular gene of interest. In IGW, 284 differentially expressed genes (DEGs) were identified in T2D donor islets compared with controls. Forty percent of DEGs showed cell-type enrichment and a large proportion significantly co-expressed with islet hormone-encoding genes; glucagon (*GCG*, 56%), amylin (*IAPP*, 52%), insulin (*INS*, 44%), and somatostatin (*SST*, 24%). Inhibition of two DEGs, *UNC5D* and *SERPINE2*, impaired glucose-stimulated insulin secretion and impacted cell survival in a human β-cell model. The exploratory use of IGW could help designing more comprehensive functional follow-up studies and serve to identify therapeutic targets in T2D.**

## Introduction

Type 2 diabetes (T2D) is a chronic condition arising from the inability of the body to maintain glucose homeostasis. It has broadly been attributed to defects in insulin secretion and increasing insulin resistance, the latter often arising because of obesity and physical inactivity (Prasad & Groop, 2015). Genome-wide association studies (GWAS) have identified 403 loci robustly associated with T2D risk (Prasad & Groop, 2015; Wood et al, 2017; Mahajan et al, 2018). A vast majority of the loci primarily influence insulin secretion (Lyssenko et al, 2008; Wood et al, 2017), thus emphasizing the central role of the islets of Langerhans and β-cell dysfunction in T2D pathogenesis.

Gene expression analysis provides a possibility to link between genetics and cellular function and is crucial for the elucidation of pathophysiological mechanisms. Information on gene expression in different tissues has greatly served the understanding of disease mechanisms. For example, the Genotype-Tissue Expression (GTEx) project (GTEx Consortium, 2013) is a pioneering example on how to share such information from deceased humans. Unfortunately, GTEx has limited information on human pancreatic islets of Langerhans, as RNA sequencing was performed on whole pancreas, not islets, and only from a limited number of deceased donors. Importantly, the pancreas needs to be removed while blood flow is still intact to retain functionality of the pancreatic cells (Jansson et al, 2016), and therefore more information on human pancreatic islets can be derived from organ donors selected for transplantation purposes where blood flow has been kept intact until pancreas excision and islet isolation is possible.

One of the main objectives of the EU-funded strategic research area, Excellence of Diabetes Research in Sweden (EXODIAB), is to facilitate diabetes research globally, for example, by creating resources and tools that can be used by the research community. One of its central platforms, The Human Tissue Laboratory, has generated a large repository of tissue samples (human pancreatic islets, fat, liver, and muscle) from deceased organ donors. It comprises gene expression (bulk RNA sequencing) and genome-

[1]Department of Clinical Sciences, Clinical Research Centre, Lund University, Malmö, Sweden   [2]Lund University Diabetes Centre (LUDC), Lund, Sweden   [3]Department of Experimental Medical Science, Developmental and Regenerative Neurobiology, Wallenberg Neuroscience Center, Lund, Sweden   [4]Stem Cells and Metabolism Research Program, Faculty of Medicine, University of Helsinki, Helsinki, Finland   [5]Department of Endocrinology, Abdominal Centre, Helsinki University Hospital, Folkhalsan Research Center, Helsinki, Finland   [6]Institute for Molecular Medicine Finland (FIMM), University of Helsinki, Helsinki, Finland   [7]Department of Immunology, Genetics and Pathology, Uppsala University, Uppsala, Sweden   [8]Department of Clinical Chemistry and Transfusion Medicine, Institute of Biomedicine, University of Gothenburg, Gothenburg, Sweden   [9]Vital-IT Group, SIB Swiss Institute of Bioinformatics, Lausanne, Switzerland   [10]Paul Langerhans Institute Dresden of the Helmholtz Center, Munich at University Hospital Carl Gustav Carus and Faculty of Medicine, TU Dresden, Dresden, Germany   [11]German Center for Diabetes Research (DZD), Munich, Germany   [12]Max Planck Institute of Molecular Cell Biology and Genetics, (MPI-CBG), Dresden, Germany   [13]Department of Clinical and Experimental Medicine, Cisanello, University Hospital, University of Pisa, Pisa, Italy   [14]Department of Cell Physiology and Metabolism, Faculty of Medicine, University of Geneva, Geneva, Switzerland   [15]Children's Hospital, Helsinki University Hospital, Helsinki, Finland   [16]Human Tissue Laboratory at Lund University Diabetes Centre, Lund, Sweden

Correspondence: rashmi.prasad@med.lu.se
*Petter Storm and Vikash Chandra contributed equally to this work.

wide genetic variation data (GWAS) from human pancreatic islets, as well as some of the target tissues for insulin. Here we describe the development of a web-based tool, the Islet Gene View (IGW), which will allow rapid and robust overview of data, as well as "look-up" of genes of interest. The underlying database shows differences in gene expression between T2D and nondiabetic donors and their relationship with specific islet phenotypes as well as the effects of genetic variation on gene expression, that is, eQTLs (Expression Quantitative Trait Loci). The tool includes some well validated examples to reinforce the usefulness and precision of the tool.

# Results

### IGW—a web resource for gene expression in human pancreatic islets

We created the IGW web tool to functionally annotate all genes expressed in human pancreatic islets and to provide a platform to look up genes of interest. IGW is accessible at https://mae.crc.med.lu.se/IsletGeneView/. IGW uses several common gene identifiers (e.g., gene symbol, Ensembl gene ID, and full gene name), and provides graphs of gene expression in relation to islet phenotypes and expression of other genes (Fig S1). An example graph is given in Fig 1. The first graph reveals gene expression in human pancreatic islets and other target tissues for insulin (e.g., a 12-donor tissue panel of biopsies from fat, liver, and skeletal muscle). The second graph shows the relationship between gene expression and purity (islet volume fraction, i.e., the proportion of endocrine component over exocrine), followed by its expression in relation to other genes expressed in the islets. Subsequent figures show gene expression in relation to glycemic status and related phenotypes such as T2D, glucose tolerance defined by HbA1c strata (normal glucose tolerance, NGT: HbA1c <6%, impaired glucose tolerance, IGT: HbA1c between 6% and 6.5%, and T2D: HbA1c > 6.5%), HbA1c as a linear variable as well as BMI, and glucose-stimulated insulin secretion (GSIS) in islets (stimulatory index, SI). Of interest is also coexpression with other islet cell-specific genes such as *INS*, *GCG*, and *IAPP*. The final graph shows islet cell-type expression from Segerstolpe et al (2016). In addition, separate tab allows researchers to download data on the top 100 co-expressed genes with the gene of interest.

### Multiple genes are differentially expressed between islets from type 2 diabetic and nondiabetic donors

In IGW, differential expression analysis of 33 donors with a clinical diagnosis of T2D and 155 diabetes-free individuals (Table S1 and Fig S1) showed that expression of 284 of a total of 14,108 genes differed significantly between the two groups (FDR ≤ 0.05, Fig 2 and Table S2). Of these differentially expressed genes (DEGs), expression of 120 genes was down-regulated, whereas that of 165 was up-regulated in T2D donor islets (Fig 2). Expression of 24 genes showed significant correlation with HbA1c levels with concordant directionality (Table S3 and Fig 2).

To validate the data that underlie the IGW, we sought to replicate our key findings by comparing our list of DEGs with previously published data. For this, we selected a study presenting DEG data from two microarray experiments (Solimena et al, 2018), run on islets from (1) partially pancreatectomized patients and (2) organ donors. Nineteen genes were replicated with directional consistency with our dataset in both the microarray sets, including *UNC5D*, *PPP1R1A*, *TMEM37*, *SLC2A2*, *ARG2*, *CAPN13*, *FFAR4*, *HHATL*, and *CHL1* (Figs 2 and 6). Expression changes in 28 genes were replicated in the expression data obtained from organ donors and that of 20 genes in the dataset of the partially pancreatectomized donors (Fig 3). Of the total 67 replicated genes, 58 also associated with HbA1c levels (Table S3). Of particular interest were the genes *HHATL*, *CHL1*, and *SLC2A2*, the expression of which was concordant to findings from a previous study including a subset of the current islets (Fadista et al, 2014). The expression of all the above three genes correlated nominally with stimulatory index (measure of GSIS in the islets) as well as with BMI. The expression of *HHATL* correlated with expression of *INS*, *SST*, and *IAPP*, *CHL1* with *GCG* and *SLC2A2* with *IAPP* (Figs S2–S4). We and others have previously shown that a genetic variant at the *CHL1* locus (rs9841287 SNP) is associated with fasting insulin concentrations (Manning et al, 2012; Scott et al, 2017). In addition, this locus was an eQTL for the *CHL1* and *CHL1-AS1* genes ($\beta$ = 0.29 and 0.27, $P$ = 0.028 and 0.04, respectively, increasing allele = G).

Of the DEGs, *GLRA1* expression was significantly associated with the stimulatory index (Fig 1). Another gene of interest was *FXYD2*, the expression of which was down-regulated in T2D donor islets and correlated with HbA1c levels. Its expression was also down-regulated in $\beta$-cells from T2D donors in a previous study using single cell RNAseq (Segerstolpe et al, 2016).

### Most genes expressed in islets and DEGs show variable expression in fat, liver, and muscle

Gene expression is a means by which a genome controls cell differentiation and consequently development of different tissues. Gene expression and its genetic regulation can be highly tissue specific or ubiquitous, the latter facilitating a higher degree of pleiotropy. The extent to which islet-expressed genes found in IGW also show expression in other tissues, especially those of relevance to T2D, is yet to be undetermined. To address this, we examined whether genes expressed in islets were also expressed in fat, liver and muscle obtained from the same individuals (defined as ≥1 cpm in >80% of samples). Indeed, we found a large proportion of genes (11,118) to be expressed in all four tissues at varying levels (Fig S5). In addition, each tissue also had unique expression patterns: 1,122 genes were expressed in islets, 713 in muscle, 529 in liver, and 1,112 in adipose tissue. Of the 284 DEGs, 36 genes showed highest expression in islets compared with the other tissues; this included the islet hormone coding *IAPP*, antisense of *SSTR5* coding *SSTR5-AS1*, *GLRA1*, *FXYD2*, and *UNC5D*. 15 genes were expressed in islet and liver including *HHATL*, whereas 40 others were expressed in at least one other tissue (Fig S5) (Fig 4). Most of the genes seemed to be ubiquitously expressed in all tissues tested (Figs S5 and 4).

### DEGs in T2D donor islets are significantly co-expressed with islet hormone encoding genes

Pancreatic islets comprise insulin-secreting $\beta$-cells, $\alpha$-cells which secrete glucagon, $\Delta$-cells which secrete somatostatin, F-cells (also

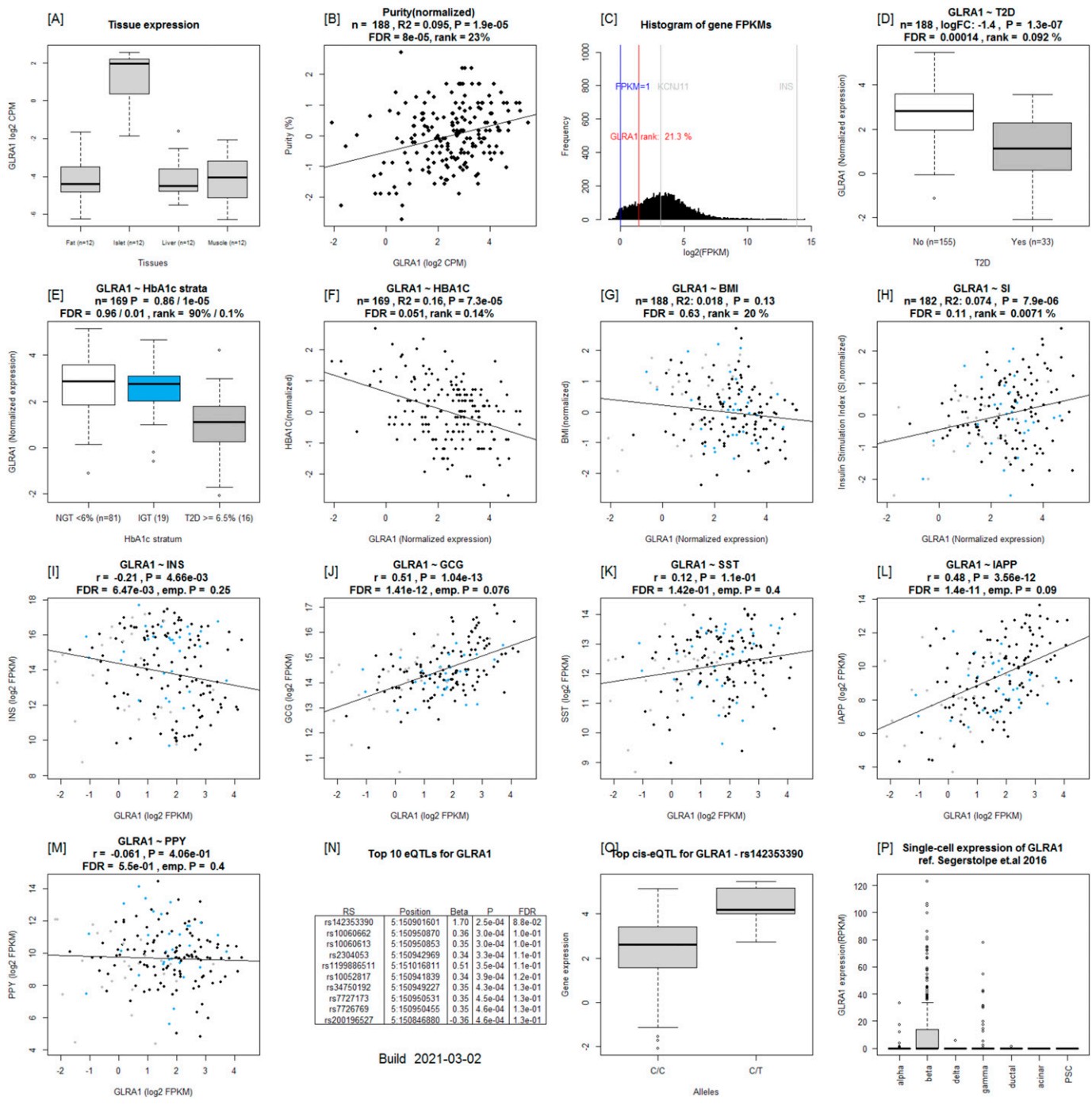

**Figure 1. Example output from Islet Gene View of *GLRA1*.**
**(A)** Expression of the gene in fat, islets, liver, and muscle in the same pool of 12 individuals. **(B)** Gene expression as a function of purity, defined as the percentage of endocrine tissue. **(C)** Expression of the selected gene in relation to other genes in islets. **(D, E, F, G, H)** (Karpichev et al, 2008): Gene expression in relation to several diabetes-related phenotypes, that is, T2D diagnosis (D), HbA1c stratum (E), continuous HbA1c (F), BMI (G), and stimulatory index (H). Test statistics are reported, namely: coefficient of determination ($R^2$), nominal *P*-value, and percentage rank among all genes as calculated based on sorted *P*-values. **(I, J, K, L, M)** (Gibson et al, 2018): Gene expression in relation to the secretory genes *INS* (I) *GCG* (J), *SST* (K), *IAPP* (L), and *PPY* (M). Spearman's *ρ* (r) and the *P*-value of the gene based on the empirical correlation distribution is reported. *INS*, insulin; *GCG*, glucagon; *SST*, somatostatin; *PPY*, pancreatic polypeptide; *IAPP*, islet amyloid polypeptide. **(N, O, P)** and (O) Top 10 eQTLs and (P) single-cell RNAseq expression data from Segerstolpe et al (2016).

called PP cells) which secrete pancreatic polypeptide Y, and ghrelin cells which produce ghrelin. In addition, β-cells also secrete islet amyloid polypeptide (*IAPP*) (also a DEG). Therefore, genes found in

IGW whose expression correlates with insulin and glucagon expression are likely to influence insulin expression or are potential downstream targets of *INS* and *GCG*. Altogether, 11,238 genes were

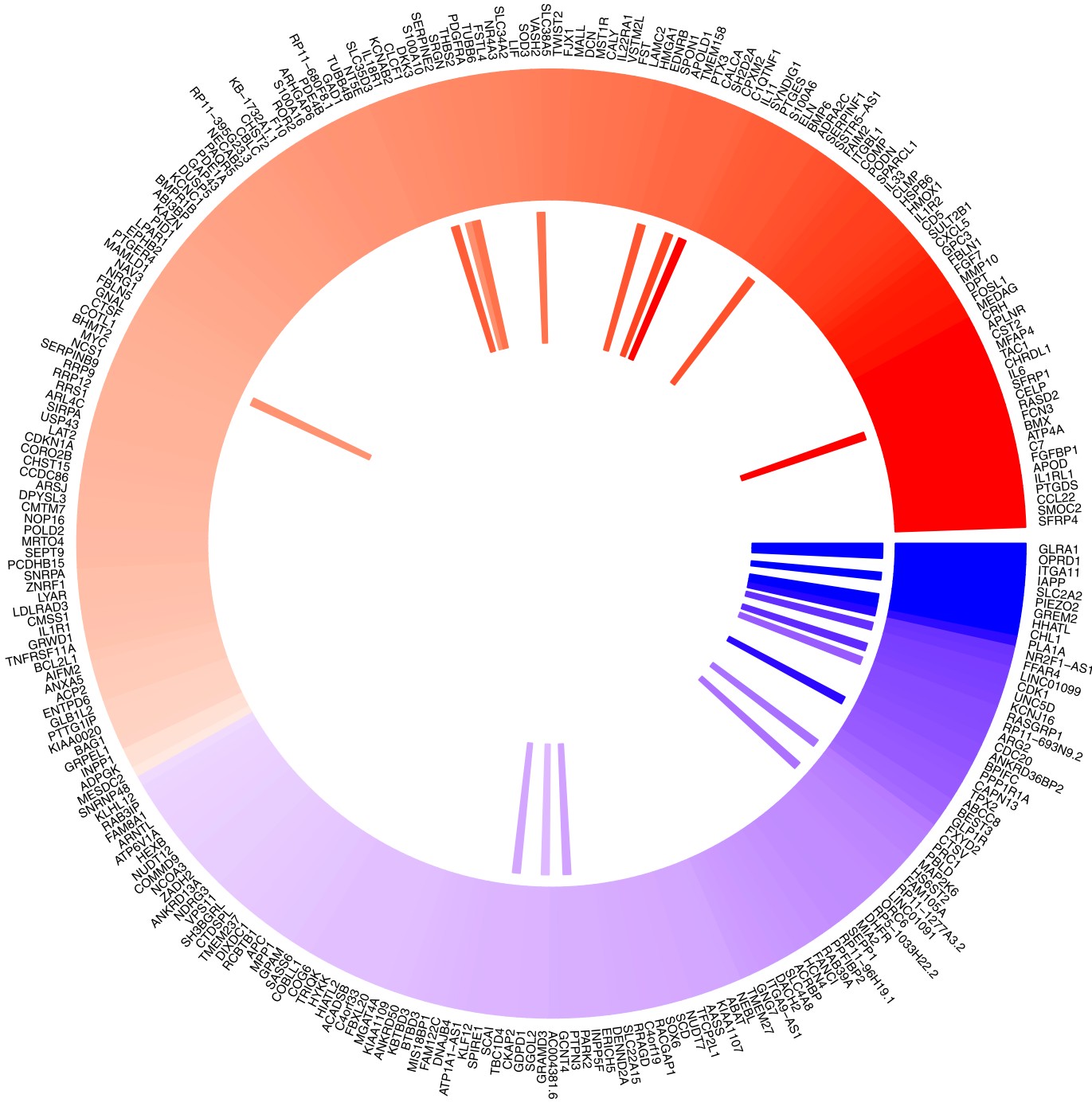

**Figure 2. Characterization of differentially expressed genes between islets from T2D donors compared with controls and association with HbA1c levels.**
Outer track: 120 genes were down-regulated (indicated in blue), whereas 164 genes were up-regulated in islets from T2D donors (indicated in red). Inner track: genes showing significant positive correlation with HbA1c levels (red) and negative correlation (blue).

co-expressed (FDR < 0.05) with *INS* and 4,519 of them were empirically significant (i.e., when gene–gene correlations for all genes expressed in islets were considered). The corresponding numbers for *GCG* were 9,347 and 789, for *SST* were 9,937 and 1,172, whereas 10,638 and 2,610 for *IAPP*. The number of genes showing positive and negative coexpression with *INS*, *GCG*, *IAPP*, and *SST* and overlapping

correlations regardless of direction are shown in Fig S6A–C, respectively. A list of top 10 genes showing the strongest correlation with islet hormone encoding genes is provided in Table S4.

The expression of 124 DEGs correlated significantly with *INS* expression of which 24 were significant at the empirical level. The corresponding numbers were 160 and 42 for *GCG*, 147 and 65 for

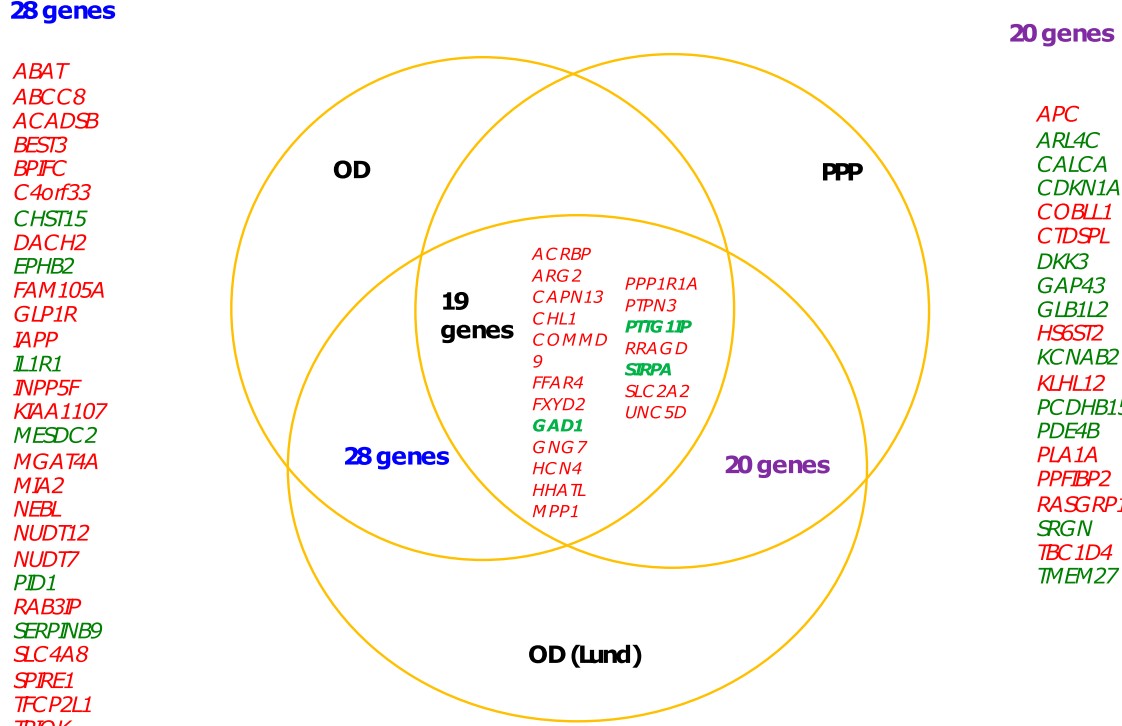

**28 genes**

*ABAT*
*ABCC8*
*ACADSB*
*BEST3*
*BPIFC*
*C4orf33*
*CHST15*
*DACH2*
*EPHB2*
*FAM105A*
*GLP1R*
*IAPP*
*IL1R1*
*INPP5F*
*KIAA1107*
*MESDC2*
*MGAT4A*
*MIA2*
*NEBL*
*NUDT12*
*NUDT7*
*PID1*
*RAB3IP*
*SERPINB9*
*SLC4A8*
*SPIRE1*
*TFCP2L1*
*TRIQK*

**20 genes**

*APC*
*ARL4C*
*CALCA*
*CDKN1A*
*COBLL1*
*CTDSPL*
*DKK3*
*GAP43*
*GLB1L2*
*HS6ST2*
*KCNAB2*
*KLHL12*
*PCDHB15*
*PDE4B*
*PLA1A*
*PPFIBP2*
*RASGRP1*
*SRGN*
*TBC1D4*
*TMEM27*

**Figure 3. Replication of differentially expressed genes in comparison with data from.**
Solimena et al (2018)Venn diagram shows the overlap of differentially expressed genes from each of the studies. OD and PPP show DE data from organ donor islets and partially pancreatectomized donor islets from Solimena et al, whereas OD (LUND) show DE genes from our data. Nine genes were replicated in all three data sets. 13 genes were replicated between the OD islets from Solimena et al (presented in blue on the left) and our data, whereas 15 genes were replicated between PPP islets and our data (presented in green on the right). The arrows indicate the direction of effect: red arrows pointing down show down-regulation in T2D islets, whereas green arrows pointing up are up-regulated.

*IAPP*, 67 and 8 for *SST*, none for *PPY*, and 0 and 2 for *GHRL* (Table S5 and Fig 5).

## Genes whose expression is altered in T2D donor islets show cellular heterogeneity in enrichment of expression

We next compared the DEG gene-set derived from IGW with a gene list of cell type–enriched genes from a previously published study applying single cell RNA sequencing (scRNAseq) of islets (Segerstolpe et al, 2016). This yielded an overlap of 20 genes enriched in α cells, 23 in β-cells, 8 in γ, 7 in Δ, 17 in acinar, and 39 in ductal cells (Fig 5). Of genes showing correlation with at least one islet hormone encoding gene, the largest proportion of genes was co-expressed with *GCG* (65%), followed by *IAPP* (57%), *INS* (50%), and *SST* (35%) (Fig 5). Two genes were co-expressed with *GHRL* (*PPP1R1A* and *FAM105A*), whereas there were no significant coexpression between the DEGs and *PPY*. The expression of 36 genes showed opposite correlation with *INS* and *GCG* (i.e., positive with *INS* and negative with *GCG*, and vice versa). These included the (i) *IAPP* and (ii) the histone acetyl transferase in the TGF β signaling pathway coding *NCOA3* genes, the expressions of both were down-regulated in T2D donor islets, and correlated positively with GCG whereas negatively with *INS* (iii), *HMGA1* (variants associated with T2D), and (iv) *BCL2L1* which promotes survival of differentiating pancreatic cells (Loo et al, 2020), both of which showed higher expression in

T2D donor islets and correlated negatively with GCG whereas positively with *INS* (Tables S1 and S5).

Several DEGs had previously been shown to be enriched in exocrine cells; of them, seven genes including *MYC*, *FST*, and *SLC38A5* in acinar cells and 17 genes, including *KCNJ16* and *CHST15* in ductal cells. The expression of these genes was also negatively correlated with purity, supporting the view of exocrine origin (Table S6). Among these DEGs, 29 genes showed higher expression in pancreatic stellate cells compared with endothelial cells (Table S7) including *SERPINE2*, *PTGDS*, and *PIEZO2*.

A pathway analysis of the DEGs showed an array of interactions between the various islet cell types involved in T2D pathogenesis (Fig 6) including genes enriched in endocrine and exocrine cells (Fig 6).

## Functional assessment of two DEGs (UNC5D and SERPINE2) by knockdown in human EndoC-βH1 cells

Based on findings in IGW, we set out to functionally validate selected findings. Expression of **UNC5D** was significantly down-regulated in islets from T2D donors and negatively correlated with HbA1c levels (Fig 7A and B). Expression of *UNC5D* correlated with that of *IAPP* (ρ = 0.57, p_emp = 0.047) and with *GCG* (ρ = 0.36, $P = 2.66 \times 10^{-07}$) expression (Fig S7). The down-regulation of *UNC5D* expression in T2D donor islets was also reported in a previous study (Solimena et al, 2018). Immunohistochemical analysis confirmed

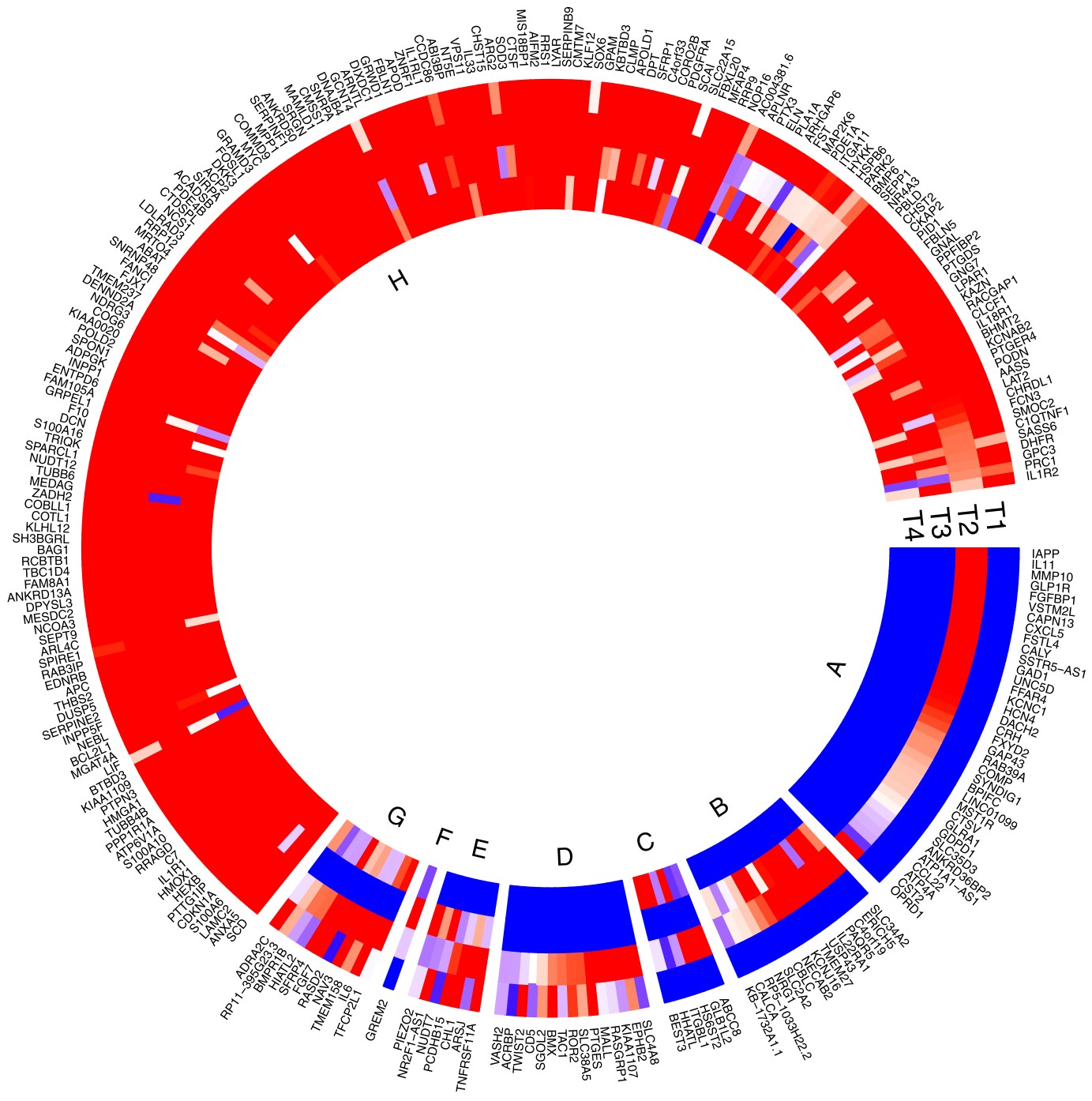

**Figure 4. Expression of the differentially expressed genes (DEGs) in fat (F), islet (I), liver (L), and muscle (M).**
T1 shows expression of DEGs in fat, T2 in islets, T3 in liver, and T4 in muscle. Expression was defined as ≥ 1 count per million (CPM). DEGs expressed in islets and not in other tissues are shown in Segment A; islet and liver in Segment B; islet and muscle in Segment C; fat and islet in Segment D; fat, islet, and liver in segment E; islet, liver, and muscle in segment F; fat islet and muscle in segment G; and all four tissues in segment H. Most of the DE genes were expressed in all four tissues; coded as blue < 1, 0 = white, red ≥ 1.

that UNC5D was expressed in islet cells, and its expression was reduced in β-cells from T2D donors (Figs 7C and S7). Moreover, scRNAseq from human pancreatic islets from our extended dataset showed that *UNC5D* is predominantly expressed in β- and Δ-cells (Figs 7D and S7).

Expression of **SERPINE2** was strongly up-regulated in T2D donor islets, and its expression was strongly positively correlated with HbA1c level (Figs 7E and F and S8). *SERPINE2* expression was nominally correlated with that of *INS* ($\rho$ = 0.17, $P$ = 0.02) and *IAPP* ($\rho$ = −0.17, $P$ = 0.019) (Fig S8). Immunostaining of SERPINE2 in adult

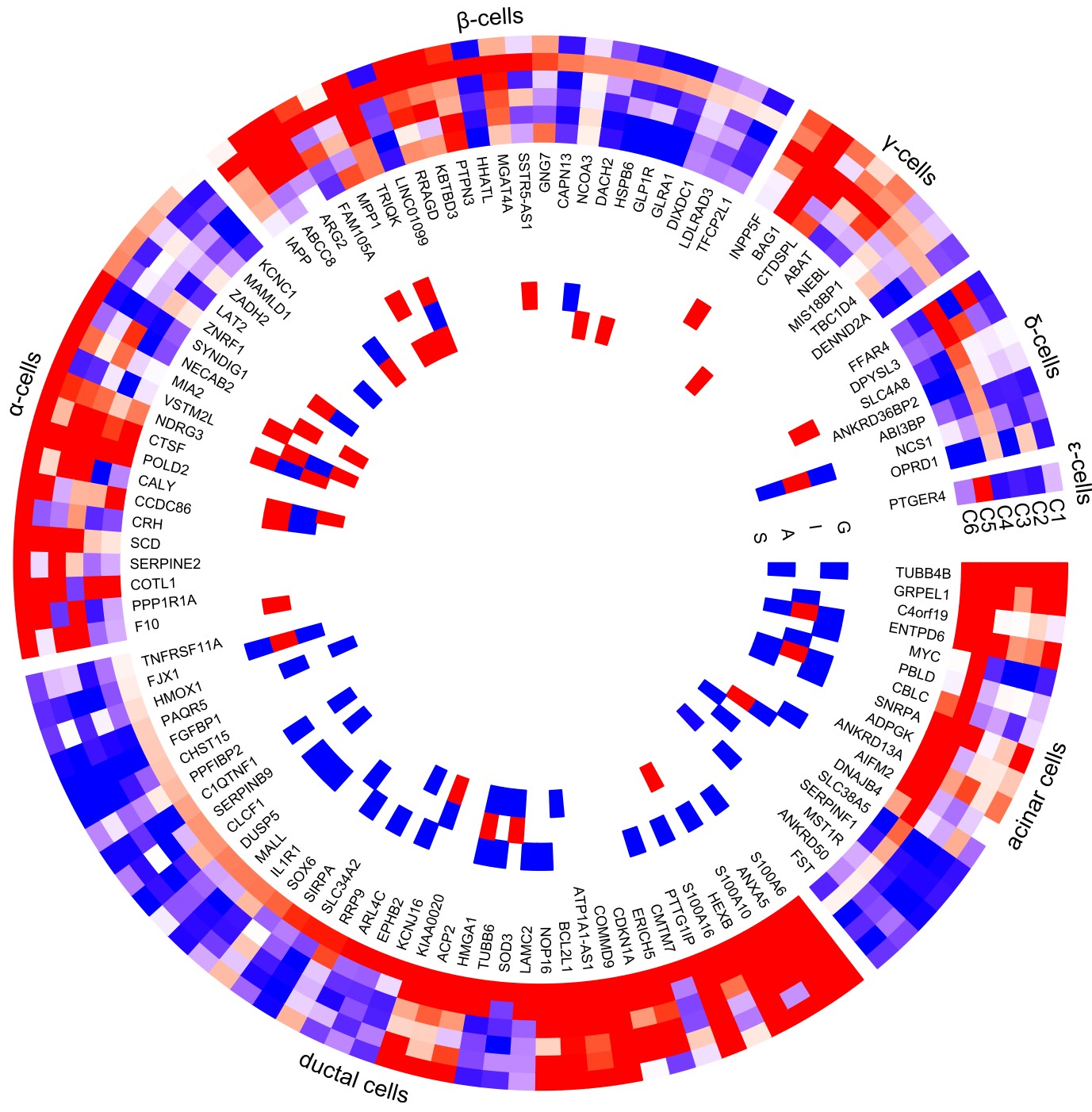

**Figure 5. Differentially expressed genes, cell type enrichment, and correlation with secretory genes.**
Differentially expressed genes which are enriched in specific islet cell types are separated into specific segments. Outer tracks show expression in scRNAseq (Segerstolpe et al, 2016). Outermost track show expression in α cells (C1), followed by β cells (C2), γ cells (C3), Δ cells (C4), acinar cells (C5), and ductal cells (C6). Mean of RPKM (log₂) values are plotted, with values code as: blue < 1 < red. Inner tracks show correlation with secretory genes starting with GCG on the outside (G), followed by INS (I), IAPP (A), and SST (S), coded as −0.5 ≥ blue, 0 = white, 0.5 ≤ red.

human pancreatic sections of normoglycemic donors showed very faint whereas strong islet-associated SERPINE2 immunoreactivity was observed in pancreatic sections of T2D donors (Fig 7G). Furthermore, scRNAseq data on a subset of islets from the same donors (Martinez-Lopez et al, unpublished) revealed that *SERPINE2*

was highly expressed in stellate cells and showed altered expression mainly in α cells from T2D donors compared with controls (cZ = 3.85) (Fig 7H).

We next assessed the functional role of *UNC5D* and *SERPINE2* using siRNA in the well-characterized human β-cell line model

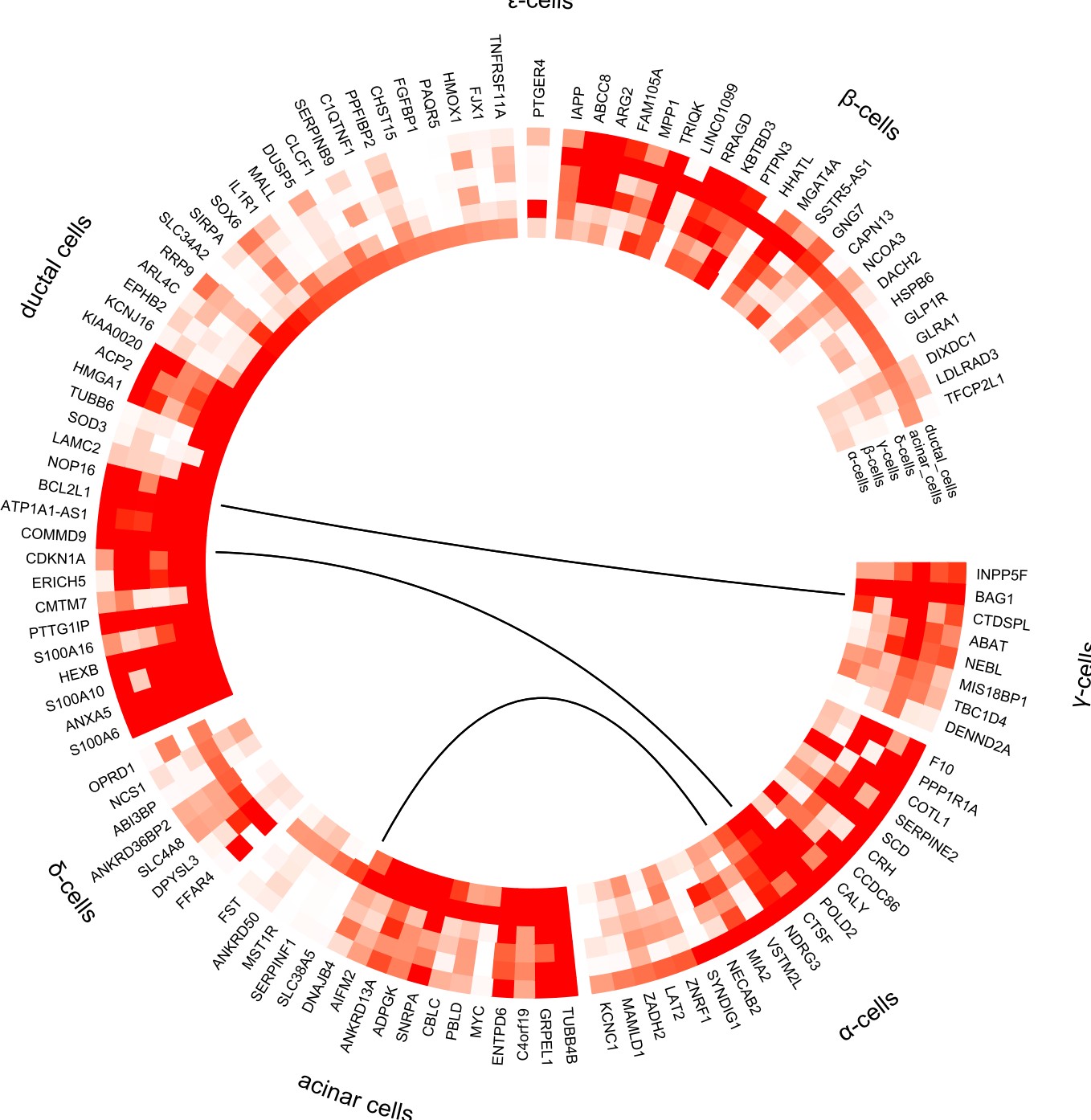

**Figure 6. Differentially expressed genes grouped by cell type enrichment as reported by Segerstolpe et al (2016).**
Differentially expressed genes which are enriched in specific islet cell types are separated into specific segments. Outer tracks show expression in scRNAseq (Segerstolpe et al, 2016). Outermost track show expression in α cells (C1), followed by β cells(C2), γ cells (C3), Δ cells (C4), acinar cells (C5), and ductal cells (C6). Mean of RPKM (log$_2$) values are plotted, with values coded as: blue < 1 < red. Inner tracks show correlation with secretory genes starting with GCG on the outside (G), followed by INS (I), IAPP (A), and SST (S), coded as −0.5 ≥ blue, 0 = white, 0.5 ≤ red. The links show networks as inferred by GeNets.

EndoC-βH1 human β-cell line (Ravassard et al, 2011). By using our previously reported reverse transfection protocol which gives >90% transfection efficiency in this cell line, we achieved a 93% ± 6% (SERPINE2) and 83% ± 4% (UNC5D) mRNA knockdown (KD) upon siRNA transfection (Fig 8A and B) (Chandra et al, 2014). KD of either SERPINE2 or UNC5D had no significant effect on total insulin content (Fig 8C). However, KD of SERPINE2 (P = 0.001) and UNC5D (P = 0.03) significantly reduced GSIS (Fig 8D) but lacked an effect on basal

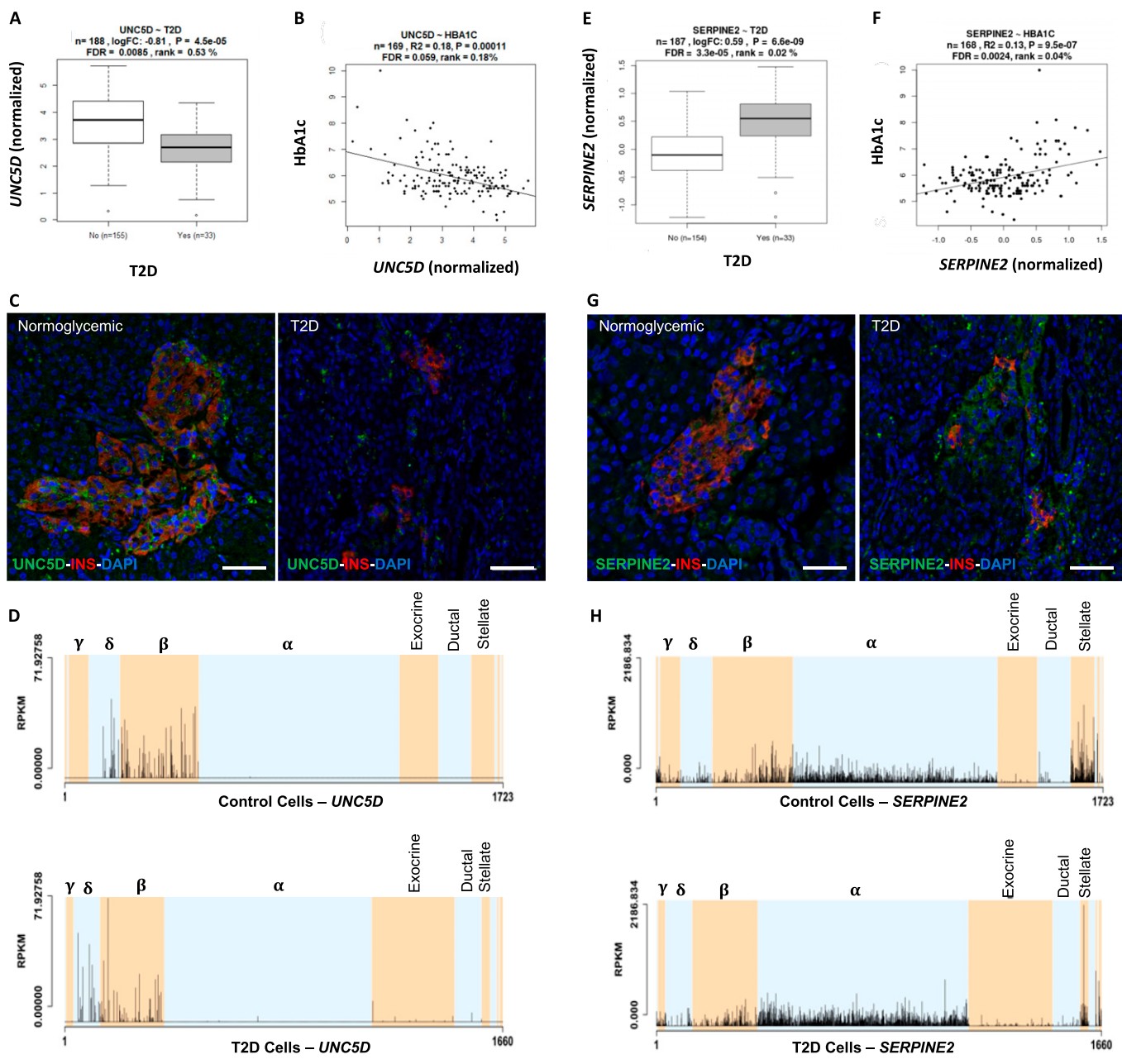

**Figure 7. SERPINE2 and UNC5D expression in islets.**
**(A, B)** UNC5D expression (A) was down-regulated in T2D donor islets and (B) correlated negatively with HbA1c levels. **(C)** Immunohistochemical staining of adult pancreas sections from normoglycemic and type 2 diabetic donors showed *UNC5D* (green) and insulin (red). Scale bar indicate 50 *μ*m, pictures were taken with a 20× objective. Nuclei are shown in blue (DAPI). **(D)** In scRNA data from the islets, *UNC5D* showed expression in Δ and *β* cells only. **(E, F)** *SERPINE2* expression was up-regulated in T2D donor islets and (F) positively correlated with that of HbA1c levels. **(G)** Immunohistochemical staining of adult pancreas sections from normoglycemic and type 2 diabetic donors showed *SERPINE2* (green) and insulin (red). *SERPINE2* expression was much higher in immunohistochemistry of the sections from T2D donors Scale bar indicate 50 *μ*m, pictures were taken with a 20× objective. Nuclei are shown in blue (DAPI). **(H)** In ScRNA data, *SERPINE2* showed ubiquitous expression, with enrichment in pancreatic stellate cells. Expression in *α* cells from T2D donors was significantly higher, whereas that in *β* cells was lower (although not statistically significant).

insulin secretion. Moreover, the stimulatory index to glucose (20G/1G) was also significantly reduced in both, by 50% ± 14% (*P* = 0.001) in *SERPINE2* KD cells and by 64% ± 014% (*P* = 0.0003) in *UNC5D* KD human *β*-cells (Fig 8E). Furthermore, we also determined insulin secretion evoked by 0.5 mM 3-isobutyl-1-methylxanthine (IBMX), a phosphodiesterase inhibitor (Siegel et al, 1980). Interestingly, IBMX-

induced insulin secretion was significantly decreased by 35 ± 6.7% (*P* = 0.0001) in *SERPINE2* KD *β*-cells, whereas we did not observe changes in *UNC5D* KD *β*-cells (Fig 8F). Finally, we studied the impact of KD of *SERPINE2* or *UNC5D* on *β*-cell survival when exposed to cytokines (IL-1*β*, TNF-*α*, and IFN-*γ*) which are implicated in inflammation in T2D (Alexandraki et al, 2006). We observed a

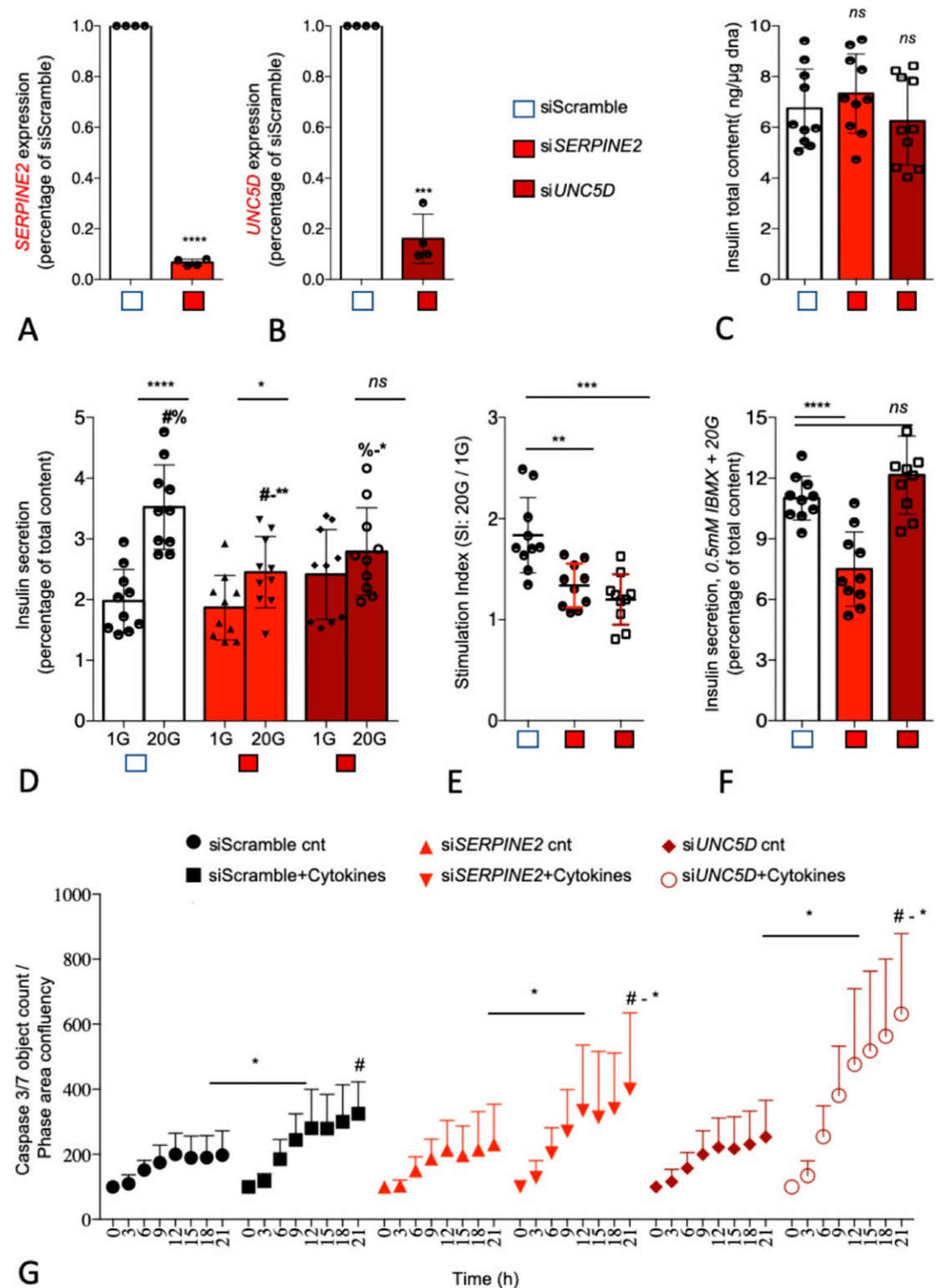

**Figure 8. SERPINE2 and UNC5D knockdown (KD) leads to impaired insulin secretion and induced apoptosis in human pancreatic EndoC-bH1 cells.**
(A, B) Effect of siRNA mediated KD of *SERPINE2* and *UNC5D* mRNA. (C) Intracellular insulin content. (D) Glucose-stimulated insulin secretion. (E) Stimulation index (ratio of high glucose 20G to low glucose 1G). (F) Insulin secretion stimulated by IBMX. (G) Effect of KD of *SERPINE2* and *UNC5D* on cytokines induced apoptosis measured with IncuCyte Caspase 3/7 green reagent. Data are shown as Mean with SD (n = 3–5). *P < 0.05, **P < 0.01, ***P < 0.001 (one-way ANOVA, followed by Tukey's test; #a, b, two-tailed unpaired *t* test).

significantly higher rate of cell death in both SERPINE2 ($P$ = 0.02) and in UNC5D ($P$ = 0.02) KD $\beta$-cells assessed by activated caspase-3/7 levels after exposure to the cytokines cocktail (Fig 8G).

### Expression quantitative traits (eQTLs)

IGW also provides information on the effect of genetic variation on gene expression in human pancreatic islets. We performed eQTL analysis using GWAS data imputed with the 1,000 Genomes Phase I reference panel and RNA data. The effect of SNPs proximal to a given gene on its expression was computed by cis-eQTL analysis using Matrix eQTL. For this analysis, only cis-eQTLs located within 2 Mb of the target genes were considered. Of all 13,169 expressed genes with eQTL data, 3,243 (24.6%) possessed at least one genome-wide significant eQTL (FDR ≤ 0.05 after adjustment for all tested variants) (Fig S9A and B). In total, there were 335,343 significant eQTLs (Fig S10).

eQTLs analysis of the DEGs harboured 1948 SNPs associated with expression of 66 genes (Table S8). Of particular interest was the prostaglandin H2 D-isomerase coding *PTGDS* gene the expression of which was up-regulated in islets from T2D donors; it significantly correlated with *INS*, GCG and *IAPP* expression and associated with HbA1c levels (Figs 2, 4, 5, 6, and S9). *PTGDS* showed a genome-wide significant eQTL signal from two linked loci which have not previously associated with T2D risk (rs28592848 and rs28375538) (Table S8 and Figs S9 and S10). Both rs28592848 and rs28375538 showed suggestive signals for association with indices of insulin secretion measures during IVGTT (acute insulin response, $P$ = 8.65 × 10$^{-4}$ and peak insulin response, $P$ = 7.40 × 10$^{-4}$) (Wood et al, 2017).

eQTLs at the *IL22RA1*, *PTGES*, *CST2*, *LINC01099*, *IL1R2*, *CKAP2*, *ROR2*, and *DHFR* loci exhibited suggestive signals for insulin secretion in the MAGIC data analysis (Table S9) (Prokopenko et al, 2014).

### Association between T2D genetic risk loci and gene expression in human pancreatic islets

Previous GWAS studies have reported 404 unique variants associated with T2D risk (Mahajan et al, 2018). These 404 T2D risk variants nominally influenced the expression of 249 target genes (eGenes). Of the 249 eGenes, 33% were associated with *INS* expression (empirical $P$ ≤ 0.05), 23% were associated with IAPP expression, 10% were associated with GCG expression, and 7% were associated with SST expression. In total, 43% of the eGenes were associated with the expression of one or more genes encoding the islet hormones. Of them, 45 (of the 249 genes) (18%) were found to be associated with T2D (FDR ≤ 0.05) (Table S10).

## Discussion

We have harnessed the potential of transcriptome sequencing and combined genome-wide genotyping in human pancreatic islets to create IGW, a unique research tool that allows the effective integration of genetic information and islet function. To this end, we used islets from 188 donors and combined in vivo and in vitro functional studies and obtained novel insights into molecular mechanisms underlying dysregulation of glucose metabolism and impaired islet function in T2D. We explored how gene expression was altered in islets from T2D donors and characterized the DEGs, for example, by studying their expression pattern in other tissues (fat, liver and muscle) from the same donors, and their coexpression with islet hormones. We also examined genetic regulation of gene expression and their link to islet function. *SERPINE2* (up-regulated in T2D) and *UNC5D* (down-regulated in T2D) genes were selected for functional analysis to explore their potential role in T2D pathogenesis. Finally, we present Islet Gene View, a web resource to visualize information on genes of interest in this comprehensive catalogue of gene expression in islets.

IGW is complementary to other related databases such as the Islet eQTL explorer (Varshney et al, 2017), and TIGER (http://tiger.bsc.es/), which connect genetic variation to expression and chromatin state, and GTEx (GTEx Consortium, 2013) which comprehensively characterizes the human transcriptome across many tissues, including whole pancreas. GTEx, however, does not provide data on the islet transcriptome, which clearly is very different from that of the total pancreas which includes the exocrine part of the pancreas. Several studies have focused on human pancreatic islets to obtain insights into general molecular mechanisms of diabetes (Fadista et al, 2014; van de Bunt et al, 2015; Varshney et al, 2017; Viñuela et al, 2019)ʼ. Our study provides another perspective by (a) exploring and characterizing DEGs and (b) by providing a web resource for visualization of gene expression which can be linked to glycemic status, expression of islet hormones and other relevant phenotypes. The platform is easily adaptable to include new data as it becomes available in-house and from other sources.

IGW identified 284 genes whose expression was altered in islets from T2D donors some of which were shown previously (Solimena et al, 2018). The variability in the transcriptomes across the data sets could partially be explained by the differences in protocols for obtaining and processing the islets, and by the method of transcriptome profiling. However, it could also be attributed to other potential sources of variation, not least, differences in proportion of endocrine component ("purity") as well as study power and population differences. In addition, the now well recognized variability in T2D phenotypes (Ahlqvist et al, 2018) may also contribute sample variability in the different studies.

Among the replicated genes were *CHL1*, *SLC2A2*, and *HHATL*, which were also reported in our previous studies based on a subset of the current islet dataset (Fadista et al, 2014). *CHL1* has also previously been shown to be associated with T2D (Belongie et al, 2017). Our data suggest that high expression of the neural adhesion molecule L1 encoding *CHL1* gene in islets could have beneficial effects on $\beta$-cell function. In support of this, *CHL1* expression was down-regulated in islets from T2D donors and correlated with insulin content (Taneera et al, 2015). The gene *SLC2A2* encoding the low affinity/high capacity GLUT2 glucose transporter protein, has previously been shown to play a key role in islet function (Thorens, 2015) and common variants in the *SLC2A2* gene are associated with T2D, insulin secretion and glycemic response to metformin therapy in recently diagnosed T2D patients (Rathmann et al, 2019). In IGW, the expression of these three genes correlated with insulin secretion in islets and with islet hormone expression. Of relevance to insulin secretion was also the gene coding for the glycine receptor

subunit α 1 protein (*GLRA1*), the expression of which was down-regulated in T2D islets, and significantly correlated with stimulatory index. *GLRA1* has been suggested to facilitate insulin secretion through a glycine-insulin autocrine feedback loop (Yan-Do et al, 2016). Moreover, *GLRA1* and *HHATL* showed higher expression in β-cells. Taken together, the data imply that these genes could serve as potential therapeutic targets in T2D.

The DEGs in IGW showed heterogeneity in expression between different islet cell populations, both endocrine and exocrine. These included genes enriched in (i) β-cells (and associated with insulin secretion e.g., *GLRA1* and *HHATL*), (ii) other endocrine cell types (α-cells, e.g., *PPP1R1A*; γ-cells, e.g., *INPP5F*; Δ-cells, e.g., *OPRD1* and *FFAR4*; ε-cells, e.g., *PTGER4*) (iii) exocrine cells (acinar cells, e.g., *MYC*; ductal cells, e.g., *CDKN1A*) (iv) pancreatic stellate cells–*SERPINE2*. Furthermore, coexpression studies might help to identify clusters of genes sharing common mechanisms for regulation. Through pathway analysis using a protein–protein interaction database as reference, we could show interaction between genes enriched both endocrine and exocrine cells. These phenomena could be interpreted as interactions of endocrine-exocrine components regulating islet function.

Genes identified by IGW showing tissue-specific expression could play a critical role in function of the specific tissues. Identification of such genes has provided molecular insights into tissue function and disease pathogenesis. To this end, we discovered DEGs that showed islet-specificity and islet-enrichment in expression compared to fat, liver and muscle from the same individual; some of these were β-cell enriched (*GLRA1*, *SSTR5-AS1*, *IAPP*, *CAPN13*, *GLP1R*, *DACH2*, and *LINC01099*), whereas others were α-cell–enriched (*CRH*, *SYNDIG1*, *CALY*, *KCNC1*, and others). A vast majority of the DEGs showed expression in all the tissues tested implying their pleiotropic nature (associated with multiple phenotypes associated) (Chavali et al, 2010; Viñuela et al, 2019). Tissue specificity and enrichment information are vital when evaluating novel therapeutic targets.

We focused more in depth on two interesting differentially expressed candidate genes in T2D islets, *UNC5D* and *SERPINE2*. Loss-of-function and gain-of-function are two possible approaches to understand the biology of a new target, as the change in the expression could be a cause or consequence of T2D progression. We chose here a loss-of-function approach with a valid human β cell line model EndoC-βH1. UNC5D is the newest member of uncoordinated-5 (*UNC5*) receptor family which acts as receptor for axon guidance factors, Netrins. Netrin factors are well described to be important for β-cell development and survival (Cirulli & Yebra, 2007; Rajasekharan & Kennedy, 2009). However, the functional role of *UNC5D* in adult human β-cells remains largely unknown. *UNC5D* is predominantly expressed in β and Δ-cells of human islets and importantly, is one of the top down-regulated genes in T2D donor islets (present dataset and scRNAseq data). Notably, none of the other genes of the UNC5 family receptors showed any differential expression pattern, whereas Netrin (*NTN1*) showed only a marginally reduced expression in the T2D donor islets (*P* = 0.01). Furthermore, we observed that *UNC5D* gene expression also showed strong positive correlation with two known T2D genes *ROBO2* and *SLC30A8* (Figs S11 and S12). It was interesting to see the strong association of *UNC5D* with the T2D target gene, *ROBO2*, which is

known to play a role in β-cell survival under different stress conditions through Slit-Robo signaling (Yang et al, 2013). *UNC5D* expression is unique to human islets with no expression in the mouse islets (Yang et al, 2011) (Fig S12). Likewise, mouse islets show a unique expression of *Unc5a*, unlike in human islets, suggestive of species-specific activity of these receptors (Fig S12). Our observations warrant an in-depth study to gain more insight into the effect of *UNC5D* on insulin secretion and β-cell survival, possibly through a Netrin-UNC5; Slit-Robo signaling loop.

*SERPINE2* (Serpin Peptidase Inhibitor, Clade E, Member 2), also known as Protease Nexin-1 (PN1) is another interesting DEG whose expression was strongly up-regulated in the T2D donor islets. To understand the biology of *SERPINE2* in human β-cells, we used RNA interference–mediated loss-of-function studies. Depletion of *SERPINE2* mRNA resulted in a markedly decreased insulin secretion in response to glucose and IBMX and increased sensitivity to inflammatory cytokines. *SERPINE2* is an extracellular matrix remodeler glycoprotein that acts as an inhibitor for trypsin, thrombin, and urokinase plasminogen activator like serine proteases (Bouton et al, 2012). Notably scRNAseq data showed *SERPINE2* to be predominantly expressed in α and stellate cells with increased expressed in T2D islets. Thus, the paracrine effect of the secreted SERPINE2 could potentially affect β-cell function and survival through its actions on the extracellular milieu. However, more detailed experiments including gain-of-function studies will be required to verify this.

GWAS have reported 403 genetic loci associated with T2D risk although the mechanisms remain largely unknown. To gain insight into mechanistic effects of SNPs, we explored whether they influence expression of unique genes (eQTLs). To this end, we identified multiple gene loci influencing gene expression in islets; the expression of these genes was altered in islets from T2D donors and correlated with insulin secretion. *PTGDS* was one such gene; it encodes an enzyme converting prostaglandin H2 to prostaglandin D2 (PGD2). High PGD2 levels have been shown to enhance insulin sensitivity (Fujitani et al, 2010), and KD of *PTGDS* results in an opposite effect resulting in insulin resistance, but also nephropathy and atherosclerosis. Whereas adipose tissue could play a role here, insulin resistance has in many studies been shown to play a key role in pathogenesis of nephropathy. Interestingly, nephropathy was primarily seen in the insulin-resistant subgroup of SIRD in the new classification of T2D (Ahlqvist et al, 2018). Although we found little support for the correlation between diabetes and *PTGDS* in islet cells in previous studies, the current results suggest that expression of the gene in islets is correlated both with diabetes status, BMI and *INS* expression, indicating that *PTGDS* function and prostaglandin levels may be connected to insulin secretion as well as peripheral insulin resistance.

Most of the results reported here can be found in the IGV. This is a web application that we have developed to visualize gene expression in simple histograms. In addition, information on purity allows for a partial separation of expression patterns in endocrine and exocrine tissue, as strong positive correlation of expression of a gene on purity is indicative of a high proportion of exocrine tissue. Relationship to glycemic status, BMI, and related phenotypes is provided as simple-to-read graphs for a specific gene.

A strength of IGW is the large sample size of islets from organ donors with maintained blood circulation, which is larger (+99

donors) than previous publications from our centre (Fadista et al, 2014; Taneera et al, 2015). Following a previous report on a smaller subset (Fadista et al, 2014), we have made several refinements to the analytical pipeline including batch correction using ComBat. We have applied refined methodology for calculating *P*-values for correlations, which are now independent of batch effects. This reduces the influence of nonspecific inter-gene correlations resulting from the normalization procedure for gene expression. All gene–gene correlations have been pre-calculated to estimate the null distribution of the correlation values. This is computationally intensive, but only has to be done once. One limitation is that the graphs and data are descriptive and cannot distinguish correlation from causation. However, we selected two genes for functional validation, demonstrating that the data can be used to further explore functionally relevant genes expressed in islets. A second caveat is the potential cell composition differences between NGT and T2D donor islets. Expression differences between T2D and NGT donors therefore can be reflective these differences in cell composition. We hope that these data, complemented with the scRNA lookups, can provide a more comprehensive picture of cell type–specific expression.

Taken together, IGV is a tool to facilitate research on human pancreatic islets and will be made accessible to the entire scientific community. The exploratory use of IGW could help designing more comprehensive functional follow-up studies and serve to identify therapeutic targets in T2D.

# Materials and Methods

### Sample acquisition

Human pancreatic islets (n = 188), fat (n = 12), liver (n = 12), and muscle (n = 12) from a mixed dataset of diabetic and nondiabetic donors were obtained through the EXODIAB network from the Nordic Network for Clinical Islet Transplantation (http://www.nordicislets.org). All procedures were approved by the ethics committee at Lund University. The isolation of total RNA including miRNA was carried out using the miRNeasy (QIAGEN) or the AllPrep DNA/RNA (QIAGEN) mini kits as described previously (Fadista et al, 2014). The quality of isolated RNA was controlled using a 2100 Bioanalyzer (Agilent Technologies) or a 2200 Tapestation (Agilent Technologies) and quantity was measured using NanoDrop 1000 (NanoDrop Technologies) or a Qubit 2.0 Fluorometer (Life Technologies). Clinical characteristics of the donors are shown in Table S1.

### Islet phenotypes

Purity of islets was assessed by dithizione staining, and estimates of the contribution of exocrine and endocrine tissue was assessed as previously described (Friberg et al, 2011).

### *Phenotypic information*

Donor characteristics are presented in Table S1. Diagnosis of Type 2 diabetes (T2D) was either based on a clinical diagnosis of T2D

(N = 33) or on an HbA1c above 6.5% (NGSP units; equal to 48 mmol/mol in IFCC) (N = 25). IGT was defined as HbA1c between 6% and 6.5% (N = 30).

Information on gender and BMI was obtained from donor records. Stimulatory index (SI) was used as a measure of GSIS. For this purpose, islets were subjected to dynamic perfusion of glucose, which was raised from 1.67 to 16.7 mmol/l for 1 h; insulin was measured at both high and low glucose. The fold change in insulin levels between the two conditions was used as a measurement of glucose-stimulated insulin secretion.

### Sample preparation for RNA sequencing

1 µg of total RNA of high quality (RIN > 8) was used for sequencing with a TruSeq RNA sample preparation kit (Illumina). We here included 99 islet samples in addition to the 89 islet samples and processed them uniformly following the same protocol as described previously (Fadista et al, 2014). Briefly, the size selection was made by Agencourt AMPure XP beads (Beckman Coulter) aiming at a fragment size over 300 bp. The resulting libraries were quality controlled on a 2100 Bioanalyzer and a 2200 Tapestation (Agilent Technologies) before combining six samples into one flow cell for sequencing on a HiSeq 2000 sequencer (Illumina).

### IGV

### *RNAseq data analysis*

The raw data were base-called and de-multiplexed using CASAVA 1.8.2 (Illumina) before alignment to hg38 with STAR version 2.4.1. To count the number of reads aligned to specific transcripts, feature-Counts (v 1.4.4) (Liao et al, 2014) was used, with GENCODE version 22 as gene, transcript and exon models. The average number of counts mapped to genes was 63.3 million reads (±35.5 million) with a median of median of 54.8M reads.

Raw data were normalized using trimmed mean of M-values (TMM) implemented in edgeR and transformed into $\log_2$ counts per million ($\log_2$ CPM), using voom (Law et al, 2014). Samples with less than 10 million reads in total were excluded from further analysis. In addition, only genes with more than 0 FPKM in 95% of samples and an average expression of more than 1 FPKM were retained, leaving 14,108 genes for analysis in the 188 samples.

A potential association between gene expression and phenotypes was analyzed by linear modeling. Voom was used to calculate variance weights, linear modeling was performed with lmFit, and *P*-values were calculated using the eBayes function in limma (Ritchie et al, 2015). *P*-values adjusted for multiple testing were calculated across all genes using Benjamini–Hochberg correction (Benjamini & Hochberg, 1995).

As expression of 50% of the genes in the dataset was correlated with purity (mostly due to admixture of exocrine tissue), we included purity as a covariate in the linear models for all association analyses, together with sex and age. Six individuals did not have T2D based upon a clinical diagnosis but based upon HbA1c above 6.5% and were excluded from the differential expression analysis of T2D versus controls.

An empirical and conservative approach was used to calculate *P*-values for gene–gene correlations. 1 million gene/gene pairs were

randomly selected from all genes after filtering for expression and the Spearman correlations for the pairs were calculated. This provided a background distribution of gene expression correlation. To calculate *P*-values from this background distribution for a given correlation, the proportion of background gene pairs with the same or more extreme correlation values was used. This provides a more robust method for detecting gene–gene pairs with high coexpression.

To investigate the biological relevance of the RNASeq and eQTL data, we investigated the relationship between previously known risk variants and gene expression in IGW. For a list of 404 variants previously associated with either T2D risk (Mahajan et al, 2018) or with β-cell function, we identified all variants nominally significant as eQTLs in the islets. The list of genes affected by eQTLs was tested for enrichment of genes whose expression was correlated with secretory genes using bootstrapping. This was carried out by repeatedly selecting random sets of genes of the same size (10,000 iterations). The mean empirical *P*-value for each run was then calculated and was used as a null distribution to calculate the probability of the observed mean *P*-value in the observed set of genes.

Pathway analysis: For the 284 genes which were differentially expressed between islets from T2D compared with normoglycemic donors, pathway analysis was performed using the InWeb (Lage et al, 2007) protein–protein interaction network implemented in GeNets (Li et al, 2018). GeNets visualization of network data were accessed at https://www.broadinstitute.org/publications/broad302496 and Intomics (https://inbio-discover.intomics.com/map.html#search).

### eQTL analysis

**Genome-wide genotying and imputation** Genotyping was performed using an Illumina OmniExpress microarray and QC was performed as previously described (Fadista et al, 2014; Viñuela et al, 2019). The resulting genotype data were phased using the SHAPEIT version 2 software, which allows for efficient phasing using an iterative expectation maximization (EM) algorithm and imputed with IMPUTE2, using the 1,000 genomes phase 1 integrated variant set as described previously.

**eQTL analysis** eQTL analysis was performed using gene expression data normalized for batch effects with ComBat, which allows for batch adjustment based on a negative binomial regression model, and the directly genotyped, as well as the imputed, genotype data. Only cis-eQTLs within 2 MB of the target gene were considered. The R package Matrix eQTL was used to analyze the association between gene expression and genetic variants.

**Islet Gene View web application** A website to access the resulting plots was developed using the Shiny web framework, software which allows for the creation of data science web applications using R code. A table of available genes can be searched and used to select a gene to investigate, which shows the corresponding set of plots for that particular gene. These plots were generated using an R script.

### Immunohistochemistry

Human pancreatic islets were processed for paraffin (6-$\mu$m sections) and cryo-embedding (10-$\mu$m sections), respectively.

Immunostaining was performed as described previously (Matsuoka et al, 2003) with the following antibodies: guinea pig $\alpha$-insulin (1:2,000; Millipore/1:800; DAKO), mouse $\alpha$-glucagon (1:2,000; Sigma-Aldrich), mouse SERPINE2 (1:2,000 [LSBio Cat. no. LS-C173926]), and goat UNC5D (1:500 [Novus Bio, Cat. no. AF1429]). Nuclear counter-staining was performed using 4,6-diamidino-2-phenylindole (DAPI, 1:6,000; Invitrogen).

### Functional studies

#### *EndoC-βH1 cells*

Human $\beta$-cell line EndoC-$\beta$H1 was obtained from Univercell Biosolution S.A.S., France (1). The cells were cultured on Matrigel (1%) and fibronectin (2 $\mu$g/ml) (Sigma-Aldrich)-coated plates in low-glucose (1 g/l) DMEM (Invitrogen) at 37°C and 5% $CO_2$ as previously described (Ravassard et al, 2011).

#### *siRNA transfection*

EndoC-$\beta$H1 cells were transfected using Lipofectamin RNAiMAX (Life Technologies) and 30 nM ON-TARGET*plus* siRNA SMARTpool for human *SERPINE2* (L-012737-00-0005; Horizon) or *UNC5D* gene (L-015286—00-0005; Horizon) or 20 nM ON-TARGET*plus* Non-targeting pool (siNT or Scramble) (D-001810-10-05; Horizon) as described (Chandra et al, 2014). Cells were harvested 96 h post-transfection for further studies.

#### *Quantitative RT-qPCR*

Total RNA was extracted from EndoC-$\beta$H1 cells using Macherey–Nagel RNA isolation kit. cDNA was prepared using the Maxima first stand cDNA synthesis kit as per manufacturers recommendations (Thermo Fisher Scientific). Briefly, 5× Hot FIREPol EvaGreen qPCR mix plus for quantitative PCR (Solis Biodyne) was used for the reactions with a Corbett Rotor-Gene 6000 (QIAGEN). The reactions were pipetted with a liquid handling system (Corbett CAS-1200; QIAGEN). All reactions were performed in duplicates on at least three biological replicates. Cyclophilin-A was used as an endogenous control. Primer sequences are available upon request.

#### *Insulin secretion and content*

EndoC-$\beta$H1 cells were transfected with siRNA on Matrigel and fibronectin-coated 24-well plates at 2 × $10^5$ cells per well. After 96 h of siRNA transfection cells were incubated in 1 mM glucose containing EndoC-$\beta$H1 culture medium for overnight and next 60 min in $\beta$KREBS (Univercell Biosolution S.A.S.) without glucose. Cells were sequentially stimulated with 500 $\mu$l $\beta$KREBS containing 1 mM glucose, 20 glucose, and then finally 20 mM glucose + 0.5 mM IBMX (#-Isobutyl-1-methylxanthine; #I5879; Sigma-Aldrich) each for 30 min at 37°C in a $CO_2$ incubator. After every incubation, the top 250 $\mu$l supernatant was carefully collected; at the end of the experiment cells were washed and lysed with TETG (Tris pH 8, Trito X-100, glycerol, NaCl, and EGTA) solution prepared as per Univercell Biosolution EndoC-$\beta$H1 manual guide for the measurement of total content. Secreted insulin after each stimulation was added to the final content to determine the % of secretion of the total content. For the measurement of total insulin content 96 h post-transfection cells were washed twice with PBS and lysed with TETG and stored in −80°C until insulin ELISA measurement. Simultaneously DNA

content was also determined to normalized the total insulin content. Secreted and intracellular insulin were measured using a commercial human insulin ELISA kit from Mercodia as per the manufacturer's instruction.

### Caspase-3/7 apoptosis assays

EndoC-$\beta$H1 cells, 2 × 10$^5$ cells per well of 24-well plate (#3526; Costar) were treated with siRNA against *SERPINE2* or *UNC5D* or Non-targeted control as described previously in siRNA methods section. After 96 h siRNA treatment, cells were washed twice with PBS and stimulated with cytokine cocktail consisting of IL-1$\beta$ (5 ng/ml; R&D Systems), and IFN-$\gamma$ (50 ng/ml; R&D Systems) and TNF-$\alpha$ (10 ng/ml; R&D Systems) in the presence of IncuCyte Caspase-3/7 green reagent (IncuCyte, #4440, 1:2,000; Essen Bioscience) for next 24 h. Every 3 h, images were taken with an IncuCyte-S3 Live-Cell Imaging system (Essen Bioscience) using 488-nm laser. Images were analyzed with IncuCyte-S3 software and apoptosis has been quantified as the ratio of green fluorescent Caspase-3/7 green active object count to phase area confluency.

### Lookups

### Indices of insulin secretion

We extracted SNPs which associated with gene expression (eQTLs) of the 284 DEGs. We performed a lookup of these SNPs in the Meta-Analyses of Glucose and Insulin-related traits Consortium (MAGIC) database, that is, searched for association of the eQTLs with indices of insulin secretion (corrected insulin response—CIR and disposition index [DI]) (Prokopenko et al, 2014).

# Data Availability

Genotype, technical and biological covariates, and sequence data have been deposited at the European Genome-phenome Archive (EGA; https://ega-archive.org/) under the following accession numbers: EGAS00001004042 [https://ega-archive.org/studies/EGAS00001004042]s; EGAS00001004044 [https://ega-archive.org/studies/EGAS00001004044], EGAS00001004056 [https://ega-archive.org/studies/EGAS00001004056].

# Supplementary Information

# Acknowledgements

Human pancreatic islets, muscle, fat, and liver samples were provided by The Nordic Network for Clinical Islet Transplantation. The work in this article has been financially supported by grants from the Swedish Research Council: strategic research environment grant (EXODIAB, 2009-1039) and project grant (2015-2558) to L Groop, networking grant (2015-06722) to RB Prasad; collaborative grant from the Swedish Foundation for Strategic Research to the Lund University Diabetes Centre (IRC 15-0067); JDRF (award 31-2008-416); Diabetes Wellness grant (720-858-16JDWG), Craoford Foundation grant (No. 20200891), Heart Lung Foundation (No. 20180522) and Hjelt Foundation grants to RB Prasad; collaborative grants with Regeneron and Eli Lilly to L Groop. This project has received funding from the Innovative Medicines Initiative 2 Joint Undertaking under grant agreement No 115881 (RHAP-SODY). This Joint Undertaking receives support from the European Union's Horizon 2020 research and innovation programme and EFPIA. This work is supported by the Swiss State Secretariat for Education' Research and Innovation (SERI) under contract number 16.0097. The opinions expressed and arguments employed herein do not necessarily reflect the official views of these funding bodies. We thank Jacqueline Postma for excellent grant management and Mattias Borell, Maria Sterner, Malin Neptin, and Malin Svensson for brilliant technical support. We also want to express our deepest gratitude to the deceased organ donors and their relatives.

## Author Contributions

O Asplund: data curation, software, formal analysis, validation, investigation, visualization, methodology, and writing—original draft.
P Storm: conceptualization, data curation, formal analysis, validation, and writing—review and editing.
V Chandra: formal analysis, validation, methodology, and writing—review and editing.
G Hatem: formal analysis, investigation, and writing—review and editing.
E Ottosson-Laakso: formal analysis and writing—review and editing.
D Mansour-Aly: investigation, methodology, and writing—review and editing.
U Krus: supervision, investigation, methodology, project administration, and writing—review and editing.
H Ibrahim: validation, investigation, methodology, and writing—review and editing.
E Ahlqvist: resources, investigation, methodology, and writing—review and editing.
T Tuomi: supervision, investigation, methodology, and writing—review and editing.
E Renström: resources, supervision, funding acquisition, investigation, methodology, project administration, and writing—review and editing.
O Korsgren: resources, funding acquisition, investigation, methodology, project administration, and writing—review and editing.
N Wierup: validation, investigation, and methodology.
M Ibberson: validation, investigation, methodology, and writing—review and editing.
M Solimena: resources, validation, investigation, methodology, and writing—review and editing.
P Marchetti: resources, supervision, validation, investigation, methodology, and writing—review and editing.
C Wollheim: supervision, investigation, methodology, and writing—review and editing.
I Artner: resources, validation, investigation, methodology, and writing—review and editing.
H Mulder: resources, supervision, investigation, methodology, and writing—review and editing.
O Hansson: conceptualization, investigation, methodology, and writing—review and editing.
T Otonkoski: conceptualization, resources, supervision, investigation, methodology, and writing—review and editing.

L Groop: conceptualization, resources, supervision, funding acquisition, methodology, project administration, and writing—review and editing.

RB Prasad: conceptualization, resources, data curation, software, formal analysis, supervision, funding acquisition, validation, investigation, visualization, methodology, project administration, and writing—original draft, review, and editing.

## Conflict of Interest Statement

The authors declare that they have no conflict of interest.

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
