## [Reviewer comments · Life Science Alliance]

Life Science Alliance

Islet Gene View - a tool to facilitate islet research

Olof Asplund, Petter Storm, Vikash Chandra, Gad Hatem, Emilia Ottosson-Laakso, Dina Mansour-Aly, Ulrika Krus, Hazem Ibrahim, Emma Ahlqvist, Tiinamajja Tuomi, Erik Renstrom, Olle Korsgren, Nils Wierup, Mark Ibberson, Michele Solimena, Piero Marchetti, Claes Wollheim, Isabella Artner, Hindrik Mulder, Ola Hansson, Timo Otonkoski, Leif Groop, and Rashmi Prasad
DOI: <https://doi.org/10.26508/Isa.202201376>

Corresponding author(s): Rashmi Prasad, Lund University

Review Timeline:

Submission Date:	2022-01-19
Editorial Decision:	2022-02-10
Revision Received:	2022-06-01
Editorial Decision:	2022-06-23
Revision Received:	2022-07-18
Accepted:	2022-07-18

Transaction Report:

February 10, 2022

Re: Life Science Alliance manuscript #LSA-2022-01376-T

Rashmi B Prasad
Lund University

Dear Dr. Prasad,

Thank you for submitting your manuscript entitled "Islet Gene View - a tool to facilitate islet research" to Life Science Alliance. The manuscript was assessed by expert reviewers, whose comments are appended to this letter. We invite you to submit a revised manuscript addressing the Reviewer comments. Please also comment on whether and how you anticipate keeping the website updated and maintained.

Thank you for this interesting contribution to Life Science Alliance. We are looking forward to receiving your revised manuscript.

Sincerely,

B. MANUSCRIPT ORGANIZATION AND FORMATTING:

Reviewer #1 (Comments to the Authors (Required)):

The paper describes a new tool for organizing gene expression results. It has some novel features that are likely to be of value. The paper seems to have two subjects and that are somewhat related. The first is the presentation of the Islet Gene View, which is a tool that probably has value to some, although there are a growing number of programs that can help with the interpretation of large quantities of data, genetic and otherwise. This approach helped define the co-expression of T2D differentially expressed genes (DEGs) and islet hormone-encoding genes, which has not been as appreciated in other studies.

The second subject is an analysis of two differentially expressed genes, UNC5D and SERPINE2. The selection of these two genes for more detailed study seem somewhat arbitrary but the knock down of these genes in the beta cell line EndoC-betaH1 provided some novel findings.

It is of note that most of the contents of this paper were made public in 2000 on BioRxiv preprint, which led to the addition of data from the group of Solimena. This seems like healthy process for strengthening a study.

Other points:

1. Some care was taken to provide references for some of the techniques, but on page 6, more could have been said about SHAPEIT, Combat and the "Shiny web framework".
2. Page 8, bottom: more should be said about "Lookups".
3. Page 10: It would be helpful to say more about why certain genes are thought to be interesting, such as HHATL, CHL1 and SLC2A2.
4. Page 12: Please say more about pancreatic stellate cells. How are they identified and why are they included in the analysis?
5. Page 16: The finding about SLC2A2 is of interest because in this study and others, it is a DEG in T2D. This finding has been puzzling because it is generally thought that the key glucose transporter in human beta cells is GLUT1 and not GLUT2. Glut2 has been shown to be important in rodent beta cells. Please comment about this interesting issue.

Reviewer #2 (Comments to the Authors (Required)):

The authors developed a web-based tool, the Islet Gene View (IGV) and describe the data acquisition and analysis in detail in the very comprehensive and well written paper. The platform is very useful as "islet organ" is often missing in RNAseq atlas platforms; The organ "pancreas" is not suitable for any extrapolation to islet transcriptomes. Furthermore, the integration and comparison of distinct analyses is very well taken.

Islet transcriptome data acquisition faces some important problems. Addressing these points in the discussion section would explain how the authors deal with this problem and sensitize the reader.

First, the comparability of RNAseq data from different subjects is questionable: in particular the comparison of RNAseq data sets of two very distinct samples: islets of NGT and T2D subjects. The authors (try to) circumvent this problem by comparing multiple distinct data sets and via the correlation between mRNAs of islet hormones and the gene of interest. But the problem remains: NGT islets have a different cell composition than T2D islets, i.e. the proportion between beta cells:alpha cells:delta cells:epsilon cells:endothelial cells:fibroblast:immune cells etc. Thus, differences in RNAseq are not necessarily due to changes within beta cells. Furthermore, the isolation procedure of islets and islet cells differentially change individual mRNA levels. Some mRNAs are more labile and faster degraded than others. Although islet tissue isolated by LCM avoids the problem of enzymatic isolation of islets, the integrity of mRNA is often poor and the islet identification is not very specific as it relies on unspecific autofluorescence of beta cells. Consequently, beta cells are recognized preferentially, however, human islets contain large amounts of alpha cells and the percentage of beta cells decreases between NGT and T2D.

The second point is the interpretation of the functional data: the authors do not explain why, both, reduced expression of UNC5D and of serpine2 affects GSIS and increases apoptosis. However, SERPINE2 mRNA levels are higher, while UNC5D is mRNA levels are lower in T2D islets compared to NGT. Is it a compensatory upregulation of Serpine2 due to defective GSIS or is the expression higher in T2D since SERPINE2 is expressed in NON-beta cells? Do these data shift the correlation (association) to causation?

Some minor points:

Fat, liver and muscle expression pattern was examined in a subset of patients (n=12) In the discussion, page 15: the statement that the samples are from the same donors. Explain more precisely whether NGT or T2D donors were used for these extra tissues and add the data to suppl. table 1. Does it matter that the donor number is low? Is the variability of RNAseq data between donors lower in fat, liver and muscle than in islets?

Page 16: add the respective protein to the gene CHL1, SLC2A2 and HHATL.
Was HHATL described before?

Page 15: correct the citations: The references are given in two adjacent brackets: (7, 27, 28) (15).

Reviewer #1 (Comments to the Authors (Required)):

The paper describes a new tool for organizing gene expression results. It has some novel features that are likely to be of value. The paper seems to have two subjects and that are somewhat related. The first is the presentation of the Islet Gene View, which is a tool that probably has value to some, although there are a growing number of programs that can help with the interpretation of large quantities of data, genetic and otherwise. This approach helped define the co-expression of T2D differentially expressed genes (DEGs) and islet hormone-encoding genes, which has not been as appreciated in other studies.

The second subject is an analysis of two differentially expressed genes, UNC5D and SERPINE2. The selection of these two genes for more detailed study seem somewhat arbitrary but the knock down of these genes in the beta cell line EndoC-betaH1 provided some novel findings.

It is of note that most of the contents of this paper were made public in 2000 on BioRxiv preprint, which led to the addition of data from the group of Solimena. This seems like healthy process for strengthening a study.

Other points:

1. Some care was taken to provide references for some of the techniques, but on page 6, more could have been said about SHAPEIT, Combat and the "Shiny web framework".

Response: Further explanations of the mentioned software have been added to the article. Please see page 6, section 2.4.2

2. Page 8, bottom: more should be said about "Lookups".

Response: Further explanation of the term "Lookup" has been added.

3. Page 10: It would be helpful to say more about why certain genes are thought to be interesting, such as HHATL, CHL1 and SLC2A2.

Response: We specifically mentioned these three genes since (a) they were also reported in our previous study based on a subset of the islets that were included in our current study and we were able to replicate this in a larger dataset (b) A second reason is elaborated in the discussion on pages 15-16, as potential biomarkers of T2D in islets and potential therapeutic targets.

4. Page 12: Please say more about pancreatic stellate cells. How are they identified and why are they included in the analysis?

Response: In Segerstolpe et al, 54 pancreatic stellate cells were identified based on high expression of collagen genes, matrix metalloproteinases, TIMP1, FN1, POSTN, and ACTA2.

Given that bulk RNA sequencing reflects expression of multiple islet cell types, and our islets had varying purity, we looked-up the expression of our DEGs in single cell RNAseq from Segerstolpe et al to see in which cell types they were expressed, and to what extent the DEGs could from the exocrine component as well.

5. Page 16: The finding about SLC2A2 is of interest because in this study and others, it is a DEG in T2D. This finding has been puzzling because it is generally thought that the key glucose transporter in human beta cells is GLUT1 and not GLUT2. Glut2 has been shown to be important in rodent beta cells. Please comment about this interesting issue.

Response: It is indeed true that GLUT1 is more highly expressed in islets and have a higher affinity for glucose at lower glucose concentrations, while GLUT2 has a lower affinity and thus requires

elevated glucose levels to be more active. However, both GLUT2 and GLUT3 have been observed in other studies to be expressed in human islets. GLUT2 may be affected by diabetes due to the differences in blood glucose levels.

Finally, we would like to thank Reviewer 1 for your very thoughtful, insightful and helpful comments.

Reviewer #2 (Comments to the Authors (Required)):

The authors developed a web-based tool, the Islet Gene View (IGW) and describe the data acquisition and analysis in detail in the very comprehensive and well written paper. The platform is very useful as "islet organ" is often missing in RNAseq atlas platforms; The organ "pancreas" is not suitable for any extrapolation to islet transcriptomes. Furthermore, the integration and comparison of distinct analyses is very well taken.

Islet transcriptome data acquisition faces some important problems. Addressing these points in the discussion section would explain how the authors deal with this problem and sensitize the reader.

First, the comparability of RNAseq data from different subjects is questionable: in particular the comparison of RNAseq data sets of two very distinct samples: islets of NGT and T2D subjects. The authors (try to) circumvent this problem by comparing multiple distinct data sets and via the correlation between mRNAs of islet hormones and the gene of interest. But the problem remains: NGT islets have a different cell composition than T2D islets, i.e. the proportion between beta cells:alpha cells:delta cells:epsilon cells:endothelial cells:fibroblast:immune cells etc. Thus, differences in RNAseq are not necessarily due to changes within beta cells. Furthermore, the isolation procedure of islets and islet cells differentially change individual mRNA levels. Some mRNAs are more labile and faster degraded than others. Although islet tissue isolated by LCM avoids the problem of enzymatic isolation of islets, the integrity of mRNA is often poor and the islet identification is not very specific as it relies on unspecific autofluorescence of beta cells. Consequently, beta cells are recognized preferentially, however, human islets contain large amounts of alpha cells and the percentage of beta cells decreases between NGT and T2D.

Response: The reviewer's point is well-taken and we have added this to the discussion (page 19). We completely agree with the reviewer that bulk RNA sequencing of islets comes with its own set of challenges, the predominant one being the variation in cell composition between the T2D and NGT donors. We also agree that the RNAseq differences do not necessarily reflect changes within beta cells per se. Isolation procedures can also contribute towards variability in expression patterns across individuals. The expression changes that are seen between T2D and NGT donors indeed reflects this change in cell composition. Secondly, there are potentially genes which are expressed ubiquitously in islet cells and have common regulatory roles across multiple cell types. Furthermore, we have included single-cell expression data to show cell type specific expression patterns. One advantage conferred by the bulk-seq is that we can also observe the specific value of looking at gene expression data across a larger number of subjects, with bulk-seq capturing inter-individual differences to a much greater extent, which is prohibitively expensive to do at the single-cell level at present.

The second point is the interpretation of the functional data: the authors do not explain why, both, reduced expression of UNC5D and of serpine2 affects GSIS and increases apoptosis. However, SERPINE2 mRNA levels are higher, while UNC5D mRNA levels are lower in T2D islets compared to NGT. Is it a compensatory upregulation of Serpine2 due to defective GSIS or is the expression

higher in T2D since SERPINE2 is expressed in NON-beta cells? Do these data shift the correlation (association) to causation?

Response:

Thank you for raising this point. We agree with the reviewer's point of view regarding SERPINE2 and hypothesize that it could be both compensatory upregulation in beta-cells or paracrine effect from non-beta-cells. Interestingly when human beta-cells (EndoC- β H1) exposed to cytokines (IL-1 β , TNF- α and IFN- γ) for 48h which are implicated in T2D inflammation, a significant induction of *SERPINE2* expression is observed (Figure R1), indicates that it could be a compensatory upregulation during progression of T2D. Also, SERPINE2 is a secreted protein (ref. Farrell DH et. al. and Baker JB et. al.), being well expressed in exocrine and alpha-cell compartment their paracrine effect on beta-cells could not be ruled out and called for a new detailed study which we think not fit in the present manuscript scope (Figure R2). Yes, we agree with the reviewer that it needs some caution while interpreting the differential expressed target genes.

Figure R1

Figure R2

Figure R1: Expression level of *SERPINE2* mRNA following 48h treatment with cytokine cocktail in EndoC- β H1 using RT-qPCR.

Figure R2: Log normalized expression level of *SERPINE2* in the adult human islet cells with sc-RNAseq (Xin et. al.)

References

Xin Y, Kim J, Okamoto H, Ni M, Wei Y, Adler C, Murphy AJ, Yancopoulos GD, Lin C, Gromada J. **RNA Sequencing of Single Human Islet Cells Reveals Type 2 Diabetes Genes.** Cell Metab. 2016 Oct 11;24(4):608-615. doi: 10.1016/j.cmet.2016.08.018. Epub 2016 Sep 22. PMID: 27667665.

Farrell DH, Wagner SL, Yuan RH, Cunningham DD. **Localization of protease nexin-1 on the fibroblast extracellular matrix.** J Cell Physiol. 1988 Feb;134(2):179-88. doi: 10.1002/jcp.1041340203. PMID: 3279057.

Baker JB, Low DA, Simmer RL, Cunningham DD. **Protease-nexin: a cellular component that links thrombin and plasminogen activator and mediates their binding to cells.** Cell. 1980 Aug;21(1):37-45. doi: 10.1016/0092-8674(80)90112-9. PMID: 6157479.

Some minor points:

Fat, liver and muscle expression pattern was examined in a subset of patients (n=12) In the discussion, page 15: the statement that the samples are from the same donors. Explain more precisely whether NGT or T2D donors were used for these extra tissues and add the data to suppl. table 1. Does it matter that the donor number is low? Is the variability of RNAseq data between donors lower in fat, liver and muscle then in islets?

Response: The donors are a mix of NGT and IGT/T2D donors. This information has been added to the manuscript. Regarding the variability of RNASeq data, all samples regardless of tissue were randomized across flow-cells to minimize experimental variability. However, given the low sample size of the other tissues in comparison with islets (12 vs 188), we cannot be totally sure if the variability comparison reflects true biology.

Page 16: add the respective protein to the gene CHL1, SLC2A2 and HHATL.
Was HHATL described before?

Response: Further descriptions of the respective proteins will be added to the manuscript. HHATL was mentioned in our previous analysis including a smaller subset of islets (Fadista et al). The current study builds upon and reports additional results. Furthermore, functions of HHATL have previously been analyzed in the context of myocardial tissue and cellular apoptosis, but has not been characterized in islets.

Page 15: correct the citations: The references are given in two adjacent brackets: (7, 27, 28) (15).

Response: the mentioned references has been corrected.

Finally, we would like to thank Reviewer 2 for your very thoughtful, insightful and helpful comments.

June 23, 2022

RE: Life Science Alliance Manuscript #LSA-2022-01376-TR

Dr. Rashmi B Prasad
Lund University
CRC, Jan Waldenströms gata 35
Malmö 20502
Sweden

Dear Dr. Prasad,

Thank you for submitting your revised manuscript entitled "Islet Gene View - a tool to facilitate islet research". We would be happy to publish your paper in Life Science Alliance pending final revisions necessary to meet our formatting guidelines.

- please upload your main and supplementary figures as single files; these will be displayed in-line in the HTML version of your paper, so please provide them as single page files (Figure 2 currently spans many pages); we do not have a limit on the number of main figures and these can be split if necessary for space
- please make sure that the author order in the manuscript and in our system match
- please consult our manuscript preparation guidelines <https://www.life-science-alliance.org/manuscript-prep> and make sure your manuscript sections are in the correct order
- please use the [10 author names, et al.] format in your references (i.e. limit the author names to the first 10)
- please add a separate section for your main figures, supplementary figures, and tables to the main manuscript text
- it seems the mention of "on behalf of the Human Tissue Laboratory at Lund University Diabetes Centre" in the author list should be accomplished by an affiliation

Figure Check:

- please update figures and legends so they are labeled A,B,C,D, etc.
- Figure S5, Figure S6 legends seem to be missing panels N-P that are in the figure
- Figure S7 legend doesn't seem to refer to all panels in the figure

A. FINAL FILES:

B. MANUSCRIPT ORGANIZATION AND FORMATTING:

Sincerely,

Reviewer #1 (Comments to the Authors (Required)):

All of my concerns have been addressed. This should be a nice addition to the literature.

Reviewer #2 (Comments to the Authors (Required)):

The authors answered all my questions. I have no further comments and suggestions.

July 18, 2022

RE: Life Science Alliance Manuscript #LSA-2022-01376-TRR

Dr. Rashmi B Prasad
Lund University
CRC, Jan Waldenströms gata 35
Malmö 20502
Sweden

Dear Dr. Prasad,

Thank you for submitting your Resource entitled "Islet Gene View - a tool to facilitate islet research". It is a pleasure to let you know that your manuscript is now accepted for publication in Life Science Alliance. Congratulations on this interesting work.

DISTRIBUTION OF MATERIALS:

Again, congratulations on a very nice paper. I hope you found the review process to be constructive and are pleased with how the manuscript was handled editorially. We look forward to future exciting submissions from your lab.

Sincerely,
